# Parameter Learning
# for Log-supermodular Distributions

**Tatiana Shpakova**
INRIA - École Normale Supérieure Paris
tatiana.shpakova@inria.fr

**Francis Bach**
INRIA - École Normale Supérieure Paris
francis.bach@inria.fr

## Abstract

We consider log-supermodular models on binary variables, which are probabilistic models with negative log-densities which are submodular. These models provide probabilistic interpretations of common combinatorial optimization tasks such as image segmentation. In this paper, we focus primarily on parameter estimation in the models from known upper-bounds on the intractable log-partition function. We show that the bound based on separable optimization on the base polytope of the submodular function is always inferior to a bound based on "perturb-and-MAP" ideas. Then, to learn parameters, given that our approximation of the log-partition function is an expectation (over our own randomization), we use a stochastic subgradient technique to maximize a lower-bound on the log-likelihood. This can also be extended to conditional maximum likelihood. We illustrate our new results in a set of experiments in binary image denoising, where we highlight the flexibility of a probabilistic model to learn with missing data.

## 1 Introduction

Submodular functions provide efficient and flexible tools for learning on discrete data. Several common combinatorial optimization tasks, such as clustering, image segmentation, or document summarization, can be achieved by the minimization or the maximization of a submodular function [1, 8, 14]. The key benefit of submodularity is the ability to model notions of diminishing returns, and the availability of exact minimization algorithms and approximate maximization algorithms with precise approximation guarantees [12].

In practice, it is not always straightforward to define an appropriate submodular function for a problem at hand. Given fully-labeled data, e.g., images and their foreground/background segmentations in image segmentation, structured-output prediction methods such as the structured-SVM may be used [18]. However, it is common (a) to have missing data, and (b) to embed submodular function minimization within a larger model. These are two situations well tackled by *probabilistic modelling*.

Log-supermodular models, with negative log-densities equal to a submodular function, are a first important step toward probabilistic modelling on discrete data with submodular functions [5]. However, it is well known that the log-partition function is intractable in such models. Several bounds have been proposed, that are accompanied with variational approximate inference [6]. These bounds are based on the submodularity of the negative log-densities. However, parameter learning (typically by maximum likelihood), which is a key feature of probabilistic modeling, has not been tackled yet. We make the following contributions:

– In Section 3, we review existing variational bounds for the log-partition function and show that the bound of [9], based on "perturb-and-MAP" ideas, formally dominates the bounds proposed by [5, 6].

– In Section 4.1, we show that for parameter learning via maximum likelihood the existing bound of [5, 6] typically leads to a degenerate solution while the one based on "perturb-and-MAP" ideas and logistic samples [9] does not.

- In Section 4.2, given that the bound based on "perturb-and-MAP" ideas is an expectation (over our own randomization), we propose to use a stochastic subgradient technique to maximize the lower-bound on the log-likelihood, which can also be extended to conditional maximum likelihood.

- In Section 5, we illustrate our new results on a set of experiments in binary image denoising, where we highlight the flexibility of a probabilistic model for learning with missing data.

## 2  Submodular functions and log-supermodular models

In this section, we review the relevant theory of submodular functions and recall typical examples of log-supermodular distributions.

### 2.1  Submodular functions

We consider submodular functions on the vertices of the hypercube $\{0,1\}^D$. This hypercube representation is equivalent to the power set of $\{1,\dots,D\}$. Indeed, we can go from a vertex of the hypercube to a set by looking at the indices of the components equal to one and from set to vertex by taking the indicator vector of the set.

For any two vertices of the hypercube, $x, y \in \{0,1\}^D$, a function $f : \{0,1\}^D \to \mathbb{R}$ is submodular if $f(x) + f(y) \geqslant f(\min\{x,y\}) + f(\max\{x,y\})$, where the min and max operations are taken component-wise and correspond to the intersection and union of the associated sets. Equivalently, the function $x \mapsto f(x + e_i) - f(x)$, where $e_i \in \mathbb{R}^D$ is the $i$-th canonical basis vector, is non-increasing. Hence, the notion of diminishing returns is often associated with submodular functions. Most widely used submodular functions are cuts, concave functions of subset cardinality, mutual information, set covers, and certain functions of eigenvalues of submatrices [1, 7]. Supermodular functions are simply negatives of submodular functions.

In this paper, we are going to use a few properties of such submodular functions (see [1, 7] and references therein). Any submodular function $f$ can be extended from $\{0,1\}^D$ to a convex function on $\mathbb{R}^D$, which is called the Lovász extension. This extension has the same value on $\{0,1\}^D$, hence we use the same notation $f$. Moreover, this function is convex and piecewise linear, which implies the existence of a polytope $B(f) \subset \mathbb{R}^D$, called the base polytope, such that for all $x \in \mathbb{R}^D$, $f(x) = \max_{s \in B(f)} x^\top s$, that is, $f$ is the support function of $B(f)$. The Lovász extension $f$ and the base polytope $B(f)$ have explicit expressions that are, however, not relevant to this paper. We will only use the fact that $f$ can be efficiently minimized on $\{0,1\}^D$, by a variety of generic algorithms, or by more efficient dedicated ones for subclasses such as graph-cuts.

### 2.2  Log-supermodular distributions

Log-supermodular models are introduced in [5] to model probability distributions on a hypercube, $x \in \{0,1\}^D$, and are defined as

$$p(x) = \frac{1}{Z(f)} \exp(-f(x)),$$

where $f : \{0,1\}^D \to \mathbb{R}$ is a submodular function such that $f(0) = 0$ and the partition function is $Z(f) = \sum_{x \in \{0,1\}^D} \exp(-f(x))$. It is more convenient to deal with the convex log-partition function $A(f) = \log Z(f) = \log \sum_{x \in \{0,1\}^D} \exp(-f(x))$. In general, the calculation of the partition function $Z(f)$ or the log-partition function $A(f)$ is intractable, as it includes simple binary Markov random fields—the exact calculation is known to be $\#P$-hard [10]. In Section 3, we review upper-bounds for the log-partition function.

### 2.3  Examples

Essentially, all submodular functions used in the minimization context can be used as negative log-densities [5, 6]. In computer vision, the most common examples are graph-cuts, which are essentially binary Markov random fields with attractive potentials, but higher-order potentials have been considered as well [11]. In our experiments, we use graph-cuts, where submodular function minimization may be performed with max-flow techniques and is thus efficient [4]. Note that there are extensions of submodular functions to continuous domains that could be considered as well [2].

# 3 Upper-bounds on the log-partition function

In this section, we review the main existing upper-bounds on the log-partition function for log-supermodular densities. These upper-bounds use several properties of submodular functions, in particular, the Lovász extension and the base polytope. Note that lower bounds based on submodular maximization aspects and superdifferentials [5] can be used to highlight the tightness of various bounds, which we present in Figure 1.

## 3.1 Base polytope relaxation with L-Field [5]

This method exploits the fact that any submodular function $f(x)$ can be lower bounded by a modular function $s(x)$, i.e., a linear function of $x \in \{0,1\}^D$ in the hypercube representation. The submodular function and its lower bound are related by $f(x) = \max_{s \in B(f)} s^\top x$, leading to:

$$A(f) = \log \sum_{x \in \{0,1\}^D} \exp\left(-f(x)\right) = \log \sum_{x \in \{0,1\}^D} \min_{s \in B(f)} \exp\left(-s^\top x\right),$$

which, by swapping the sum and min, is less than

$$\min_{s \in B(f)} \log \sum_{x \in \{0,1\}^D} \exp\left(-s^\top x\right) = \min_{s \in B(f)} \sum_{d=1}^D \log\left(1 + e^{-s_d}\right) \stackrel{\text{def}}{=} A_{\text{L-field}}(f). \quad (1)$$

Since the polytope $B(f)$ is tractable (through its membership oracle or by maximizing linear functions efficiently), the bound $A_{\text{L-field}}(f)$ is tractable, i.e., computable in polynomial time. Moreover, it has a nice interpretation through convex duality as the logistic function $\log(1 + e^{-s_d})$ may be represented as $\max_{\mu_d \in [0,1]} -\mu_d s_d - \mu_d \log \mu_d - (1 - \mu_d) \log(1 - \mu_d)$, leading to:

$$A_{\text{L-field}}(f) = \min_{s \in B(f)} \max_{\mu \in [0,1]^D} -\mu^\top s + H(\mu) = \max_{\mu \in [0,1]^D} H(\mu) - f(\mu),$$

where $H(\mu) = -\sum_{d=1}^D \left\{\mu_d \log \mu_d + (1 - \mu_d) \log(1 - \mu_d)\right\}$. This shows in particular the convexity of $f \mapsto A_{\text{L-field}}(f)$. Finally, [6] shows the remarkable result that the minimizer $s \in B(f)$ may be obtained by minimizing a simpler function on $B(f)$, namely the squared Euclidean norm, thus leading to algorithms such as the minimum-norm-point algorithm [7].

## 3.2 "Pertub-and-MAP" with logistic distributions

Estimating the log-partition function can be done through optimization using "pertub-and-MAP" ideas. The main idea is to perturb the log-density, find the maximum a-posteriori configuration (i.e., perform optimization), and then average over several random perturbations [9, 17, 19].

The Gumbel distribution on $\mathbb{R}$, whose cumulative distribution function is $F(z) = \exp(-\exp(-(z + c)))$, where $c$ is the Euler constant, is particularly useful. Indeed, if $\{g(y)\}_{y \in \{0,1\}^D}$ is a collection of independent random variables $g(y)$ indexed by $y \in \{0,1\}^D$, each following the Gumbel distribution, then the random variable $\max_{y \in \{0,1\}^D} g(y) - f(y)$ is such that we have $\log Z(f) = \mathbb{E}_g\left[\max_{y \in \{0,1\}^D} \{g(y) - f(y)\}\right]$ [9, Lemma 1]. The main problem is that we need $2^D$ such variables, and a key contribution of [9] is to show that if we consider a factored collection $\{g_d(y_d)\}_{y_d \in \{0,1\}, d=1,\ldots,D}$ of i.i.d. Gumbel variables, then we get an upper-bound on the log partition-function, that is, $\log Z(f) \leq \mathbb{E}_g \max_{y \in \{0,1\}^D} \left\{\sum_{d=1}^D g_d(y_d) - f(y)\right\}$.

Writing $g_d(y_d) = [g_d(1) - g_d(0)]y_d + g_d(0)$ and using the fact that (a) $g_d(0)$ has zero expectation and (b) the difference between two independent Gumbel distributions has a logistic distribution (with cumulative distribution function $z \mapsto (1 + e^{-z})^{-1}$) [15], we get the following upper-bound:

$$A_{\text{Logistic}}(f) = \mathbb{E}_{z_1,\ldots,z_D \sim \text{logistic}}\left[\max_{y \in \{0,1\}^D} \{z^\top y - f(y)\}\right], \quad (2)$$

where the random vector $z \in \mathbb{R}^D$ consists of independent elements taken from the logistic distribution. This is always an upper-bound on $A(f)$ and it uses only the fact that submodular functions are efficient to optimize. It is convex in $f$ as an expectation of a maximum of affine functions of $f$.

## 3.3 Comparison of bounds

In this section, we show that $A_{\text{L-field}}(f)$ is always dominated by $A_{\text{Logistic}}(f)$. This is complemented by another result within the maximum likelihood framework in Section 4.

**Proposition 1.** *For any submodular function $f : \{0,1\}^D \to \mathbb{R}$, we have:*

$$A(f) \leqslant A_{\text{Logistic}}(f) \leqslant A_{\text{L-field}}(f). \qquad (3)$$

*Proof.* The first inequality was shown by [9]. For the second inequality, we have:

$$
\begin{aligned}
A_{\text{Logistic}}(f) =\;& \mathbb{E}_z\big[\max_{y \in \{0,1\}^D} z^\top y - f(y)\big] \\
=\;& \mathbb{E}_z\big[\max_{y \in \{0,1\}^D} z^\top y - \max_{s \in B(f)} s^\top y\big] \text{ from properties of the base polytope } B(f), \\
=\;& \mathbb{E}_z\big[\max_{y \in \{0,1\}^D} \min_{s \in B(f)} z^\top y - s^\top y\big], \\
=\;& \mathbb{E}_z\big[\min_{s \in B(f)} \max_{y \in \{0,1\}^D} z^\top y - s^\top y\big] \text{ by convex duality}, \\
\leqslant\;& \min_{s \in B(f)} \mathbb{E}_z\big[\max_{y \in \{0,1\}^D} (z - s)^\top y\big] \text{ by swapping expectation and minimization}, \\
=\;& \min_{s \in B(f)} \mathbb{E}_z\big[\textstyle\sum_{d=1}^{D}(z_d - s_d)_+\big] \text{ by explicit maximization}, \\
=\;& \min_{s \in B(f)} \big[\textstyle\sum_{d=1}^{D} \mathbb{E}_{z_d}(z_d - s_d)_+\big] \text{ by using linearity of expectation}, \\
=\;& \min_{s \in B(f)} \big[\textstyle\sum_{d=1}^{D} \int_{-\infty}^{+\infty}(z_d - s_d)_+ P(z_d) dz_d\big] \text{ by definition of expectation}, \\
=\;& \min_{s \in B(f)} \big[\textstyle\sum_{d=1}^{D} \int_{s_d}^{+\infty}(z_d - s_d)\frac{e^{-z_d}}{(1+e^{-z_d})^2} dz_d\big] \text{ by substituting the density function}, \\
=\;& \min_{s \in B(f)} \textstyle\sum_{d=1}^{D} \log(1 + e^{-s_d}), \text{ which leads to the desired result.} \qquad \square
\end{aligned}
$$

In the inequality above, since the logistic distribution has full support, there cannot be equality. However, if the base polytope is such that, with high probability $\forall d, |s_d| \geq |z_d|$, then the two bounds are close. Since the logistic distribution is concentrated around zero, we have equality when $|s_d|$ is large for all $d$ and $s \in B(f)$.

**Running-time complexity of $A_{\text{L-field}}$ and $A_{\text{logistic}}$.** The logistic bound $A_{\text{logistic}}$ can be computed if there is efficient MAP-solver for submodular functions (plus a modular term). In this case, the divide-and-conquer algorithm can be applied for L-Field [5]. Thus, the complexity is dedicated to the minimization of $O(|V|)$ problems. Meanwhile, for the method based on logistic samples, it is necessary to solve $M$ optimization problems. In our empirical bound comparison (next paragraph), the running time was the same for both methods. Note however that for parameter learning, we need a *single* SFM problem per gradient iteration (and not $M$).

**Empirical comparison of $A_{\text{L-field}}$ and $A_{\text{logistic}}$.** We compare the upper-bounds on the log-partition function $A_{\text{L-field}}$ and $A_{\text{logistic}}$, with the setup used by [5]. We thus consider data from a Gaussian mixture model with 2 clusters in $\mathbb{R}^2$. The centers are sampled from $\mathcal{N}([3,3], I)$ and $\mathcal{N}([-3,-3], I)$, respectively. Then we sampled $n = 50$ points for each cluster. Further, these $2n$ points are used as nodes in a complete weighted graph, where the weight between points $x$ and $y$ is equal to $e^{-c||x-y||}$.

We consider the graph cut function associated to this weighted graph, which defines a log-supermodular distribution. We then consider conditional distributions, one for each $k = 1, \ldots, n$, on the events that at least $k$ points from the first cluster lie on the one side of the cut and at least $k$ points from the second cluster lie on the other side of the cut. For each conditional distribution, we evaluate and compare the two upper bounds. We also add the tree-reweighted belief propagation upper bound [23] and the superdifferential-based lower bound [5].

In Figure 1, we show various bounds on $A(f)$ as functions of the number on conditioned pairs. The logistic upper bound is obtained using 100 logistic samples: the logistic upper-bound $A_{\text{logistic}}$ is close to the superdifferential lower bound from [5] and is indeed significantly lower than the bound $A_{\text{L-field}}$. However, the tree-reweighted belief propagation bound behaves a bit better in the second case, but its calculation takes more time, and it cannot be applied for general submodular functions.

### 3.4 From bounds to approximate inference

Since linear functions are submodular functions, given any convex upper-bound on the log-partition function, we may derive an approximate marginal probability for each $x_d \in \{0, 1\}$. Indeed, following [9], we consider an exponential family model $p(x|t) = \exp(-f(x) + t^\top x - A(f - t))$, where

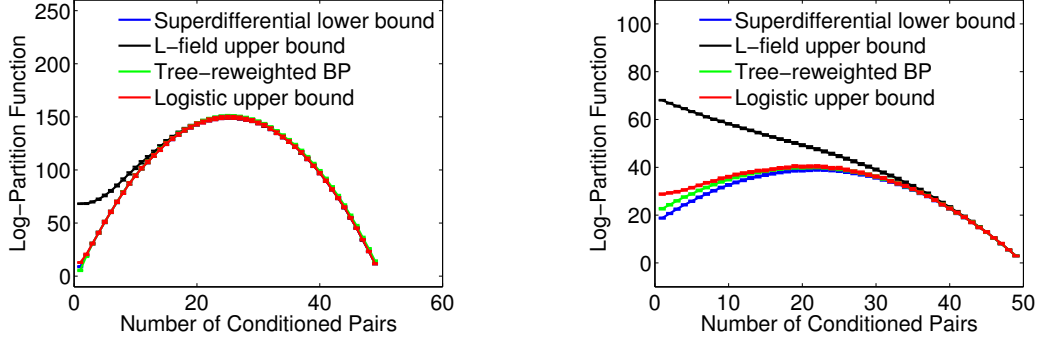

(a) Mean bounds with confidence intervals, $c = 1$.  (b) Mean bounds with confidence intervals, $c = 3$.

Figure 1: Comparison of log-partition function bounds for different values of $c$. See text for details.

$f - t$ is the function $x \mapsto f(x) - t^\top x$. When $f$ is assumed to be fixed, this can be seen as an exponential family with the base measure $\exp(-f(x))$, sufficient statistics $x$, and $A(f - t)$ is the log-partition function. It is known that the expectation of the sufficient statistics under the exponential family model $\mathbb{E}_{p(x|t)} x$ is the gradient of the log-partition function [23]. Hence, any approximation of this log-partition gives an approximation of this expectation, which in our situation is the vector of marginal probabilities that an element is equal to $1$.

For the L-field bound, at $t = 0$, we have $\partial_{t_d} A_{\text{L-field}}(f - t) = \sigma(s_d^*)$, where $s^*$ is the minimizer of $\sum_{d=1}^{D} \log(1 + e^{-s_d})$, thus recovering the interpretation of [6] from another point of view.

For the logistic bound, this is the inference mechanism from [9], with $\partial_{t_d} A_{\text{logistic}}(f - t) = \mathbb{E}_z y^*(z)$, where $y^*(z)$ is the maximizer of $\max_{y \in \{0,1\}^D} z^\top y - f(y)$. In practice, in order to perform approximate inference, we only sample $M$ logistic variables. We could do the same for parameter learning, but a much more efficient alternative, based on mixing sampling and convex optimization, is presented in the next section.

## 4 Parameter learning through maximum likelihood

An advantage of log-supermodular probabilistic models is the opportunity to learn the model parameters from data using the maximum-likelihood principle. In this section, we consider that we are given $N$ observations $x_1, \dots, x_N \in \{0, 1\}^D$, e.g., binary images such as shown in Figure 2.

We consider a submodular function $f(x)$ represented as $f(x) = \sum_{k=1}^{K} \alpha_k f_k(x) - t^\top x$. The modular term $t^\top x$ is explicitly taken into account with $t \in \mathbb{R}^D$, and $K$ base submodular functions are assumed to be given with $\alpha \in \mathbb{R}_+^K$ so that the function $f$ remains submodular. Assuming the data $x_1, \dots, x_N$ are independent and identically (i.i.d.) distributed, then maximum likelihood is equivalent to minimizing:

$$\min_{\alpha \in \mathbb{R}_+^K,\, t \in \mathbb{R}^D} -\frac{1}{N} \sum_{n=1}^{N} \log p(x_n | \alpha, t) = \min_{\alpha \in \mathbb{R}_+^K,\, t \in \mathbb{R}^D} \frac{1}{N} \sum_{n=1}^{N} \Big\{ \sum_{k=1}^{K} \alpha_k f_k(x_n) - t^\top x_n + A(f) \Big\},$$

which takes the particularly simple form

$$\min_{\alpha \in \mathbb{R}_+^K,\, t \in \mathbb{R}^D} \sum_{k=1}^{K} \alpha_k \Big( \frac{1}{N} \sum_{n=1}^{N} f_k(x_n) \Big) - t^\top \Big( \frac{1}{N} \sum_{n=1}^{N} x_n \Big) + A(\alpha, t), \tag{4}$$

where we use the notation $A(\alpha, t) = A(f)$. We now consider replacing the intractable log-partition function by its approximations defined in Section 3.

### 4.1 Learning with the L-field approximation

In this section, we show that if we replace $A(f)$ by $A_{\text{L-field}}(f)$, we obtain a degenerate solution. Indeed, we have

$$A_{\text{L-field}}(\alpha, t) = \min_{s \in B(f)} \sum_{d=1}^{D} \log(1 + e^{-s_d}) = \min_{s \in B(\sum_{k=1}^{K} \alpha_K f_K)} \sum_{d=1}^{D} \log(1 + e^{-s_d + t_d}).$$

This implies that Eq. (4) becomes

$$\min_{\alpha\in\mathbb{R}_+^K,\, t\in\mathbb{R}^D}\ \min_{s\in B(\sum_{k=1}^K \alpha_K f_K)} \sum_{k=1}^K \alpha_k\Big(\frac{1}{N}\sum_{n=1}^N f_k(x_n)\Big) - t^\top\Big(\frac{1}{N}\sum_{n=1}^N x_n\Big) + \sum_{d=1}^D \log\big(1+e^{-s_d+t_d}\big).$$

The minimum with respect to $t_d$ may be performed in closed form with $t_d - s_d = \log\frac{\langle x\rangle_d}{1-\langle x\rangle_d}$, where $\langle x\rangle = \frac{1}{N}\sum_{n=1}^N x_n$. Putting this back into the equation above, we get the equivalent problem:

$$\min_{\alpha\in\mathbb{R}_+^K}\ \min_{s\in B(\sum_{k=1}^K \alpha_K f_K)} \sum_{k=1}^K \alpha_k\Big(\frac{1}{N}\sum_{n=1}^N f_k(x_n)\Big) - s^\top\Big(\frac{1}{N}\sum_{n=1}^N x_n\Big) + \text{ const },$$

which is equivalent to, using the representation of $f$ as the support function of $B(f)$:

$$\min_{\alpha\in\mathbb{R}_+^K}\sum_{k=1}^K \alpha_k\big[\tfrac{1}{N}\sum_{n=1}^N f_k(x_n) - f_k\big(\tfrac{1}{N}\sum_{n=1}^N x_n\big)\big].$$

Since $f_k$ is convex, by Jensen's inequality, the linear term in $\alpha_k$ is non-negative; thus maximum likelihood through L-field will lead to a degenerate solution where all $\alpha$'s are equal to zero.

## 4.2 Learning with the logistic approximation with stochastic gradients

In this section we consider the problem (4) and replace $A(f)$ by $A_{\text{Logistic}}(f)$:

$$\min_{\alpha\in\mathbb{R}_+^K,\, t\in\mathbb{R}^D} \sum_{k=1}^K \alpha_k\langle f_k(x)\rangle_{\text{emp.}} - t^\top\langle x\rangle_{\text{emp.}} + \mathbb{E}_{z\sim\text{logistic}}\Big[\max_{y\in\{0,1\}^D} z^\top y + t^\top y - \sum_{k=1}^K \alpha_k f(y)\Big],\quad (5)$$

where $\langle M(x)\rangle_{\text{emp.}}$ denotes the empirical average of $M(x)$ (over the data).

Denoting by $y^*(z,t,\alpha)\in\{0,1\}^D$ the maximizers of $z^\top y + t^\top y - \sum_{k=1}^K \alpha_k f(y)$, the objective function may be written:

$$\sum_{k=1}^K \alpha_k\big[\langle f_k(x)\rangle_{\text{emp.}} - \langle f_k(y^*(z,t,\alpha))\rangle_{\text{logistic}}\big] - t^\top\big[\langle x\rangle_{\text{emp.}} - \langle y^*(z,t,\alpha)\rangle_{\text{logistic}}\big] + \langle z^\top y^*(z,t,\alpha)\rangle_{\text{logistic}}.$$

This implies that at optimum, for $\alpha_k > 0$, then $\langle f_k(x)\rangle_{\text{emp.}} = \langle f_k(y^*(z,t,\alpha))\rangle_{\text{logistic}}$, while, $\langle x\rangle_{\text{emp.}} = \langle y^*(z,t,\alpha)\rangle_{\text{logistic}}$, the expected values of the sufficient statistics match between the data and the optimizers used for the logistic approximation [9].

In order to minimize the expectation in Eq. (5), we propose to use the projected stochastic gradient method, not on the data as usually done, but on our own internal randomization. The algorithm then becomes, once we add a weighted $\ell_2$-regularization $\Omega(t,\alpha)$:

- **Input**: functions $f_k$, $k = 1,\ldots,K$, and expected sufficient statistics $\langle f_k(x)\rangle_{\text{emp.}}\in\mathbb{R}$ and $\langle x\rangle_{\text{emp.}}\in[0,1]^D$, regularizer $\Omega(t,\alpha)$.
- **Initialization**: $\alpha = 0, t = 0$
- **Iterations**: for $h$ from 1 to $H$
  - Sample $z\in\mathbb{R}^D$ as independent logistics
  - Compute $y^* = y^*(z,t,\alpha)\in\arg\max_{y\in\{0,1\}^D} z^\top y + t^\top y - \sum_{k=1}^K \alpha_k f(y)$
  - Replace $t$ by $t - \frac{C}{\sqrt{h}}\big[y^* - \langle x\rangle_{\text{emp.}} + \partial_t\Omega(t,\alpha)\big]$
  - Replace $\alpha_k$ by $\big(\alpha_k - \frac{C}{\sqrt{h}}\big[\langle f_k(x)\rangle_{\text{emp.}} - f_k(y^*) + \partial_{\alpha_k}\Omega(t,\alpha)\big]\big)_+$.
- **Output**: $(\alpha, t)$.

Since our cost function is convex and Lipschitz-continuous, the averaged iterates are converging to the global optimum [16] at rate $1/\sqrt{H}$ (for function values).

## 4.3 Extension to conditional maximum likelihood

In experiments in Section 5, we consider a joint model over two binary vectors $x, z\in\mathbb{R}^D$, as follows

$$p(x,z|\alpha,t,\pi) = p(x|\alpha,t)p(z|x,\pi) = \exp(-f(x) - A(f))\prod_{d=1}^D \pi_d^{\delta(z_d\neq x_d)}(1-\pi_d)^{\delta(z_d=x_d)},\quad (6)$$

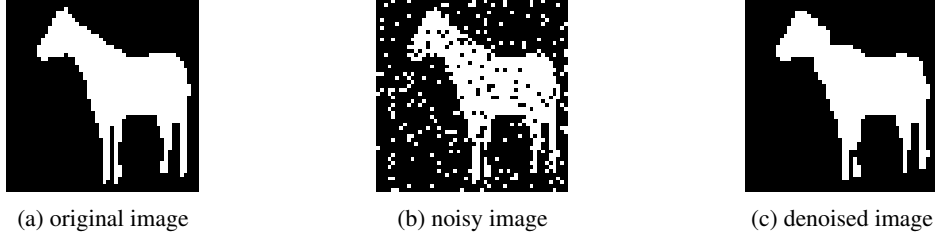

<div align="center">

(a) original image      (b) noisy image      (c) denoised image

Figure 2: Denoising of a horse image from the Weizmann horse database [3].

</div>

which corresponds to sampling $x$ from a log-supermodular model and considering $z$ that switches the values of $x$ with probability $\pi_d$ for each $d$, that is, a noisy observation of $x$. We have:

$$\log p(x, z|\alpha, t, \pi) \;\; = \;\; -f(x) - A(f) + \textstyle\sum_{d=1}^{D} \big\{ -\log(1 + e^{u_d}) + x_d u_d + z_d u_d - 2x_d z_d u_d \big\},$$

with $u_d = \log \frac{\pi_d}{1 - \pi_d}$ which is equivalent to $\pi_d = (1 + e^{-u_d})^{-1}$.

Using Bayes rule, we have $p(x|z, \alpha, t, \pi) \propto \exp(-f(x) - A(f) + x^\top u - 2x^\top(u \circ z))$, which leads to the log-supermodular model $p(x|z, \alpha, t, \pi) = \exp(-f(x) + x^\top(u - 2u \circ z) - A(f - u + 2u \circ z))$.

Thus, if we observe both $z$ and $x$, we can consider a conditional maximization of the log-likelihood (still a convex optimization problem), which we do in our experiments for supervised image denoising, where we assume we know both noisy and original images at training time. Stochastic gradient on the logistic samples can then be used. Note that our conditional ML estimation can be seen as a form of approximate conditional random fields [13].

While supervised learning can be achieved by other techniques such as structured-output-SVMs [18, 20, 22], our approach also applies when we do not observe the original image, which we now consider.

### 4.4 Missing data through maximum likelihood

In the model in Eq. (6), we now assume we only observed the noisy output $z$, and we perform parameter learning for $\alpha, t, \pi$. This is a latent variable model for which maximum likelihood can be readily applied. We have:

$$
\begin{aligned}
\log p(z|\alpha, t, \pi) \;\; &= \;\; \log \textstyle\sum_{x \in \{0,1\}} p(z, x|\alpha, t, \pi) \\
&= \;\; \log \textstyle\sum_{x \in \{0,1\}^D} \exp(-f(x) - A(f)) \prod_{d=1}^{D} \pi_d^{\delta(z_d \neq x_d)} (1 - \pi_d)^{\delta(z_d = x_d)} \\
&= \;\; A(f - u + 2u \circ z) - A(f) + z^\top u - \textstyle\sum_{d=1}^{D} \log(1 + e^{u_d}).
\end{aligned}
$$

In practice, we will assume that the noise probability $\pi$ (and hence $u$) is uniform across all elements. While we could use majorization-minimization approaches such as the expectation-minimization algorithm (EM), we consider instead stochastic subgradient descent to learn the model parameters $\alpha, t$ and $u$ (now a non-convex optimization problem, for which we still observed good convergence).

## 5 Experiments

The aim of our experiments is to demonstrate the ability of our approach to remove noise in binary images, following the experimental set-up of [9]. We consider the training sample of $N_{train} = 100$ images of size $D = 50 \times 50$, and the test sample of $N_{test} = 100$ binary images, containing a horse silhouette from the Weizmann horse database [3]. At first we add some noise by flipping pixels values independently with probability $\pi$. In Figure 2, we provide an example from the test sample: the original, the noisy and the denoised image (by our algorithm).

We consider the model from Section 4.3, with the two functions $f_1(x), f_2(x)$ which are horizontal and vertical cut functions with binary weights respectively, together with a modular term of dimension $D$. To perform minimization we use graph-cuts [4] as we deal with positive or attractive potentials.

**Supervised image denoising.** We assume that we observe $N = 100$ pairs $(x_i, z_i)$ of original-noisy images, $i = 1, \dots, N$. We perform parameter inference by maximum likelihood using stochastic subgradient descent (over the logistic samples), with regularization by the squared $\ell_2$-norm, one

| noise $\pi$ | max-marg. | std | mean-marginals | std | SVM-Struct | std |
|---|---|---|---|---|---|---|
| 1% | 0.4% | <0.1% | 0.4% | <0.1% | 0.6% | <0.1% |
| 5% | 1.1% | <0.1% | 1.1% | <0.1% | 1.5% | <0.1% |
| 10% | 2.1% | <0.1% | 2.0% | <0.1% | 2.8% | 0.3% |
| 20% | 4.2% | <0.1% | 4.1% | <0.1% | 6.0% | 0.6% |

Table 1: Supervised denoising results.

| | $\pi$ is fixed | | | | $\pi$ is not fixed | | | |
|---|---|---|---|---|---|---|---|---|
| $\pi$ | max-marg. | std | mean-marg. | std | max-marg. | std | mean-marg. | std |
| 1% | 0.5% | <0.1% | 0.5% | <0.1% | 1.0% | - | 1.0% | - |
| 5% | 0.9% | 0.1% | 1.0% | 0.1% | 3.5% | 0.9% | 3.6% | 0.8% |
| 10% | 1.9% | 0.4% | 2.1% | 0.4% | 6.8% | 2.2% | 7.0% | 2.0% |
| 20% | 5.3% | 2.0% | 6.0% | 2.0% | 20.0% | - | 20.0% | - |

Table 2: Unsupervised denoising results.

parameter for $t$, one for $\alpha$, both learned by cross-validation. Given our estimates, we may denoise a new image by computing the "max-marginal", e.g., the maximum a posteriori $\max_x p(x|z, \alpha, t)$ through a single graph-cut, or computing "mean-marginals" with 100 logistic samples. To calculate the error we use the normalized Hamming distance and 100 test images.

Results are presented in Table 1, where we compare the two types of decoding, as well as a structured output SVM (SVM-Struct [22]) applied to the same problem. Results are reported in proportion of correct pixels. We see that the probabilistic models here slightly outperform the max-margin formulation[1] and that using mean-marginals (which is optimal given our loss measure) lead to slightly better performance.

**Unsupervised image denoising.** We now only consider $N = 100$ noisy images $z_1, \ldots, z_N$ to learn the model, without the original images, and we use the latent model from Section 4.4. We apply stochastic subgradient descent for the difference of the two convex functions $A_{\text{logistic}}$ to learn the model parameters and use fixed regularization parameters equal to $10^{-2}$.

We consider two situations, with a known noise-level $\pi$ or with learning it together with $\alpha$ and $t$. The error was calculated using either max-marginals and mean-marginals. Note that here, structured-output SVMs cannot be used because there is no supervision. Results are reported in Table 2. One explanation for a better performance for max-marginals in this case is that the unsupervised approach tends to oversmooth the outcome and max-marginals correct this a bit.

When the noise level is known, the performance compared to supervised learning is not degraded much, showing the ability of the probabilistic models to perform parameter estimation with missing data. When the noise level is unknown and learned as well, results are worse, still better than a trivial answer for moderate levels of noise (5% and 10%) but not better than outputting the noisy image for extreme levels (1% and 20%). In challenging fully unsupervised case the standard deviation is up to 2.2% (which shows that our results are statistically significant).

# 6 Conclusion

In this paper, we have presented how approximate inference based on stochastic gradient and "perturb-and-MAP" ideas could be used to learn parameters of log-supermodular models, allowing to benefit from the versatility of probabilistic modelling, in particular in terms of parameter estimation with missing data. While our experiments have focused on simple binary image denoising, exploring larger-scale applications in computer vision (such as done by [24, 21]) should also show the benefits of mixing probabilistic modelling and submodular functions.

**Acknowledgements.** We acknowledge support the European Union's H2020 Framework Programme (H2020-MSCA-ITN-2014) under grant agreement n°642685 MacSeNet, and thank Sesh Kumar, Anastasia Podosinnikova and Anton Osokin for interesting discussions related to this work.

## Footnotes

[1][9] shows a stronger difference, which we believe (after consulting with authors) is due to lack of convergence for the iterative algorithm solving the max-margin formulation.

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
