[Reviews · NeurIPS 2016]

Reviewer 1

Summary

Authors show that the two bounds on the log-partition function from [5] and [9], respectively, can be ranked (ineq. (3)). The latter superior bound then is used for model parameter estimation by matching the sufficient statistics (eqn. line 189/190), utilizing submodular inference as a subroutine. Minimization can be done by interleaving MAP-perturb with updating the model parameters (due to [15]), which makes the approach efficient. Experiments illustrate the contribution.

Qualitative Assessment

This is a nice paper. Exploiting the MAP-perturb approach for model parameter learning appears to be new and should have an impact to the field, as it essentially relies on established inference methods. Authors should address the influence of the number of logistic samples: how to choose a small but sufficiently large number?

Confidence in this Review

2-Confident (read it all; understood it all reasonably well)


Reviewer 2

Summary

This work considers the class of probabilistic models whose log-densities are supermodular. For such models, [5] and [9] give upper-bounds "L-Field" and "Logistic", respectively, on the intractable log-partition normalizer. This paper first shows that Logistic is always a tighter upper bound than L-Field. It then goes on to show that L-Field is also not useful for parameter learning (gives degenerate solution setting all parameters to 0), but that Logistic gives an expression for log-likelihood that can be optimized via projected stochastic gradient. Finally, it shows that for the problem of image de-noising, even with just the noisy images available, parameter learning is possible.

Qualitative Assessment

Technical quality: The math seems correct, though in general showing a few more intermediate steps in derivations would make the work of the reader easier (can push the proof to the supplement to make room). In the proof of Proposition 1 for instance, it would be nice to have more detail on how the final equality is obtained. The experiments compare to an SVM baseline, but not to the parameter learning done in [9]. Specifically, [9] does parameter learning for the spin glass model and shows that it achieves an error of 1.8% compared to an SVM's 8.2%. What is different about the learning done in this work (besides the use of a different probabilistic model)? Novelty/originality: Section 4.2 is the main contribution of this work, but it does not seem like a very large delta on the work of [9]. Plugging the A_{Logistic} approximation for the partition function into the log likelihood directly yields the learning algorithm. Potential impact or usefulness: The ability to estimate model parameters even without the original (non-noisy) images is useful. However, experiments on more realistic data (a variety of more complex images for instance, rather than a single horse silhouette) would be better evidence of this usefulness. Clarity and presentation: The paper is generally well-written, though, as mentioned above, a few more intermediate steps in some of the derivations would be helpful. Additional minor notes below. Minor notes: Figure 1: It's hard to see the superdifferential lower bound line at all here. Maybe state in the figure caption that it is entirely covered by the logistic upper bound's line. Experiments: The paper states "using mean marginals (which is optimal given our loss measure) lead to slightly better performance". This is true or the supervised results (Table 1), but the opposite is true for the unsupervised results (Table 2). Is there a logical explanation for why max-marginals does better in the unsupervised setting? Line 92: "The method exploits" -> "This method exploits" Line 115: "log Z" -> "log Z(f)" Line 270: "allowing to benefit" -> "allowing us to benefit"

Confidence in this Review

2-Confident (read it all; understood it all reasonably well)


Reviewer 3

Summary

This paper proposes a more tighter upper bound of log-partition function based on "Perturb-and-MAP" compared with L-field for log-supermodular models. L-field aims to find modular function (decoupling variables like mean field) to upper bound log-partition function, which leads to a convex problem and the bound is tractable and computable in polynomial time. However, when maximizing likelihood, L-filed always results in degenerative solutions. In contrast, L-logistic upper bound provided in this paper can avoid this problem. Moreover, the gradient of MLE usually contains expectation operation which is intractable, this paper proposes to use stochastic gradient over logistic random sample z, which gives unbiased estimation.

Qualitative Assessment

Strength: [1] L-logistic is tighter than L-field, and avoid leading to degenerate solutions. This is quite nice. Weakness: [1] L-field can be computed in polynomial time, while L-logistic have to do expectation operation over logistic random variable. This might be much slower? And, it might be hard to decide how many samples is sufficient? Also, the number of samples might affect the tightness of the upper bound. Would it be nice to compare L-field and L-logistic vs. number of samples? Comment: [1] Using stochastic gradient over logistic random variable to avoid the expectation operation in the gradient is a cute idea. But, how much we can benefit from this? Any quantitative comparison might be nice? [2] The trend of max-marginal and mean-marginal are not consistent between table 1&2? [3] Comparison of L-field and L-logistic are implemented on synthetic/toy data, any experiments on real datasets?

Confidence in this Review

2-Confident (read it all; understood it all reasonably well)


Reviewer 4

Summary

This paper shows how to learn parameters of log-supermodular models with approximate inference based on stochastic gradient and perturb-and-MAP. Experiments on image denoising demonstrate that the proposed approach would be useful for parameters estimation with missing data.

Qualitative Assessment

This paper shows how to learn parameters of log-supermodular models with approximate inference based on stochastic gradient and perturb-and-MAP. Experiments on image denoising demonstrate that the proposed approach would be useful for parameters estimation with missing data. There are several problems with this paper. 1. Experiments are weak and the justifications of baseline are missing. It is hard to tell the benefits of the proposed algorithm from the experiments, although I understand this submission is a theoretic-focus paper, and learning the parameters of log-supermodular models would be useful for readers who are also working on this line. 1) I am not sure if there is a correct baseline in the experiments. In the supervised denoising experiments, why not using a denoising algorithm involved similar approximate inference (e.g. perturb-and-map random fields, or mean-field approximate inference)? It is also hard to tell which reasons bring the improvements of the proposed approach over SVM-struct. I would like to see the justification about the baseline in the rebuttal. 2) Table 1 and Table 2 do not provide sufficient explanations. Table 1 shows inconsistent results with Table 2 for the comparison between max-marginals and mean-marginals, I am not sure why. It would be better to have this explanation somewhere. 2. It is not clear to see why the maximum likelihood through L-field lead to a degenerate solution. It seems that detailed proofs and references are missing in line 183-184.

Confidence in this Review

2-Confident (read it all; understood it all reasonably well)


Reviewer 5

Summary

The submitted paper addresses parameter learning of log-supermodular distributions via an approximation for the partition function based on the perturb-and-MAP idea. The authors show that the proposed approximation is always tigther than L-Field and evaluate their approach in synthetic experiments, as well as an image denoising application.

Qualitative Assessment

I enjoyed reading this well written paper and also like the taken approach as well as the obtained results. My main criticism is that the title of the paper "promises" more than what is actually presented in the paper (i.e. the title suggests, in my opinion, a very general treatment of the learning problem, while the experiments are limited to rather easy cases of learning --- and it is hard to tell from the paper how well the proposed approach will work on more involved learning problems). Some comments: * I would really appreciate if you could evaluate your proposed bounds in a bit more detail, e.g. more synthetic experiments. How does the bound change as a function of the number of logistic samples? * I suggest to add a brief section comparing the the computational complexity of your approach with that of Djolonga & Krause. * The paper suggests that L-Field is not suitable for learning log-supermodular distributions but the authors only provide very limited results themselves. Did you do any more involved experiments in which for example you tried to learn "the inner life" of a submodular function, e.g. the parameterization of facility location functions or the cut functions you used for denoising? If so, what did you observe? * Do you have an explanation for your observations in lines 263ff? Please add standard-deviations to your results in tables 1 & 2 to make them more meaningful. * Did you try any other reasonable learning approaches for comparison, e.g. ML learning using sampling to estimate gradients? * Can you provide some insights into how the likelihood of the original problem changes during the learning process? (by approximating the likelihood; alternatively, how does a lower-bound on the likelihood change?) Minor comments: * Lines 6-7 in the abstract: People not very familiar with the topic probably cannot make sense of this. * I suggest to cite some general overview paper on submodularity in machine learning by J. Bilmes or A. Krause in paragraph 1 of the introduction. * Line 52: set min/max in math-style? * Line 68: subclasses instead of subcases? * I would shift the sentence in lines 88-90 to the experimental secction (I found it confusing, where it is). * Line 112: Refer specifically to Lemma 1 in [9]? * Eq. 2: Add brackets * Line 184: wth => will * Please provide a bit more detail on the cut-functions mentioned in Lines 239 (what are the weights you used? Binary?)

Confidence in this Review

2-Confident (read it all; understood it all reasonably well)


Reviewer 6

Summary

The article focusses on parameter learning with potentially missing data and applies perturb-and-MAP approaches to yield a tighter bound for approximate inference. The experiments have been conducted on a binary labelling problem, but in general the method is applicable to submodular functions/log supermodular models.

Qualitative Assessment

The article was well written regarding grammar and language. Having troubles understanding the theoretical contributions of this paper, I might recommend to be very clear in the results section. It didn't become obvious what is your method and if you are comparing to existing literature. A few more questions might arise from my lack of understanding, could be motivation for further elaboration: How drastic in performance is the difference between the method using L-Field bounds and Perturb-and-MAP bounds on the example of image denoising? How well is the approximated inference with your method to in comparison to the global optimum obtained by the graph cut algorithm (which should be exact for the submodular energy on the binary problem)? Regarding the outer form, I'd recommend not to place many long formulas within the text itself which obfuscates understanding.

Confidence in this Review

1-Less confident (might not have understood significant parts)